# Extracellular Vesicle MicroRNAs in Heart Failure: Pathophysiological Mediators and Therapeutic Targets

**DOI:** 10.3390/cells12172145

**Published:** 2023-08-25

**Authors:** Changhai Tian, Jessica N. Ziegler, Irving H. Zucker

**Affiliations:** 1Department of Toxicology and Cancer Biology, University of Kentucky, Lexington, KY 40536, USA; zieglerj1@xavier.edu; 2Department of Cellular and Integrative Physiology, University of Nebraska Medical Center, Omaha, NE 68198, USA; izucker@unmc.edu

**Keywords:** extracellular vesicles, microRNAs, sorting mechanism, biomarkers, heart failure

## Abstract

Extracellular vesicles (EVs) are emerging mediators of intracellular and inter-organ communications in cardiovascular diseases (CVDs), especially in the pathogenesis of heart failure through the transference of EV-containing bioactive substances. microRNAs (miRNAs) are contained in EV cargo and are involved in the progression of heart failure. Over the past several years, a growing body of evidence has suggested that the biogenesis of miRNAs and EVs is tightly regulated, and the sorting of miRNAs into EVs is highly selective and tightly controlled. Extracellular miRNAs, particularly circulating EV-miRNAs, have shown promising potential as prognostic and diagnostic biomarkers for heart failure and as therapeutic targets. In this review, we summarize the latest progress concerning the role of EV-miRNAs in HF and their application in a therapeutic strategy development for heart failure.

## 1. Introduction

Cardiovascular disease, including chronic heart failure (HF), is the number one cause of mortality and morbidity in the United States. The 2021 statistical update from the American Heart Association estimated more than 6 million Americans are living with HF. This equates to approximately 1.8% of the total US population, and this number will rise by 46% over the next several years to reach approximately 8 million by 2030 [1]. Accumulating evidence suggests that HF is a vicious cycle and neurohormonal dysregulation, including sympathetic nervous system excitation, the renin-angiotensin system activation, and others, contribute to the pathogenesis of HF [2,3,4,5,6,7,8].

It has also been well documented that clinically relevant signaling pathways (e.g., those activating angiotensin and adrenergic pathways) are well-coordinated during the development and progression of HF through a signature pattern of miRNA expression highly associated with cardiac hypertrophy and fibrosis in both mouse and human HF [9,10]. Importantly, these studies suggest that individual cardiac stress-inducible miRNAs, such as miR-208 and miR-195, are sufficient to provoke HF.

miRNAs are a class of noncoding RNAs and are well-known as fundamental regulators of gene expression by binding the 3′ UTR of targeting mRNAs or other regions, including the 5′ UTR of mRNA, coding sequence, and gene promoters to induce mRNA degradation or translational repression [11,12,13,14]. Despite the involvement of miRNAs in the pathophysiology of maladaptive remodeling in HF [15,16,17,18], extracellular miRNAs, especially those carried in extracellular vesicles (EVs), are promising biomarkers and paracrine mediators of intra and inter-organ communication as well as potential therapeutic targets. This review will focus on the roles of EV miRNAs in intercellular and inter-organ communication in the progression of HF and summarize the current progress in miRNA selectively packaged into EVs. We also focus on utilizing and targeting extracellular miRNAs as diagnostic biomarkers and therapy in HF.

## 2. miRNA Biogenesis and Extracellular Vesicle Selection

miRNAs, as a dominating class of small ncRNAs, are approximately 22 nucleotides in length and are produced by Drosha and Dicer, two RNase III proteins (see Figure 1).

miRNAs mediate RNA silencing by targeting most protein-coding transcripts to modulate pathological processes associated with HF. Extracellular vesicles are natural membrane-bound nanoparticles released by all cells, have a wide range of diameters, and are produced by different intracellular pathways. A growing body of literature shows that EVs are important mediators of intercellular and intra-organ communication in the pathogenesis of HF by transferring bioactive materials, including miRNAs. To better understand the various roles of EV miRNAs in the progression of HF, it will be necessary to understand the biogenesis of miRNAs and EVs and the selection mechanisms of extracellular miRNAs.

### 2.1. miRNAs and Extracellular Vesicle Biogenesis

miRNA biogenesis is tightly controlled at all levels, including transcription, processing, modification, and Argonaute (AGO) protein loading as well as RNA decay [22]. Increasing evidence supports the view that dysregulation of miRNA biogenesis is associated with human HF [23,24,25,26]. miRNA biogenesis can be regulated either by the canonical pathway (Figure 1-I) or non-canonical pathways (Figure 1-II, III, V). The canonical pathway comprises a series of steps: First, miRNA transcription is carried out by RNA polymerase II (Pol II) and Pol II-associated transcription factors and epigenetic regulators to pri-miRNAs [27,28,29,30], followed by nuclear processing of pri-miRNAs by Drosha and DGCR8 (microprocessor complex) to pre-miRNAs [31,32]. Second, following nuclear events, nuclear export of pre-miRNAs is mediated by a transport complex comprising exportin 5 (XPO5) and GTP-binding RAN (Ras-related nuclear protein) in the cytoplasm [33], where pre-miRNAs are processed by Dicer, one RNase III-type enzyme, and TRBP (HIV-1 transactivating response (TAR) RNA-binding protein) to miRNA-duplex [34,35]. Third, selective miRNA-duplexes are loaded onto AGO proteins to form an RNA-induced silencing complex (RISC), unwinding the miRNA-duplexes. Theoretically, if one miRNA strand is selectively loaded onto an AGO protein to form the mature RISC, which further cleaves target mRNA or represses mRNA translation by binding to the 3′-UTR of mRNA [36,37], the other strand will be ejected from the RISC and subjected to degradation. However, increasing evidence not only suggests that miRNAs can bind to other regions of targets in addition to 3′-UTR including 5′-UTR, coding area, and promoter regions to activate translation or regulate transcription [38,39,40] but also both miRNA strands were found to functionally co-exist and participate in the pathogenesis of HF [41,42].

In addition to the canonical pathway, there are several non-canonical pathways involved in miRNA biogenesis. These non-canonical pathways are generally divided into two groups: Drosha/DGCR8-independent (Figure 1-II, III) and Dicer-independent pathways (Figure 1-V). Mirtron is one type of pre-miRNA produced from introns of mRNAs by splicing [43]. The 7-methylguanosine (m^7^G)-capped pre-miRNAs are also produced by Drosha/DGCR8-independent pathway [44]. Both pre-miRNAs are still dependent on Dicer-mediated cytoplasmic maturation, but they differ in nuclear export. Mirtron is exported by XPO5, whereas m7G-capped pre-miRNA is exported by XPO1 [22,44]. In addition, some miRNAs are made through Dicer-independent pathways (e.g., miR-451). The product of pri-miRNA-451 cleavage by Drosha is too short to be processed by Dicer, and pre-miRNA-451 will be directly loaded onto AGO2 and then sliced into an AGO-cleaved pre-miRNA-451 (ac-pre-miR-451), which will be further matured by poly (A)-specific ribonuclease PARN-mediated trimming [45].

Extracellular vesicles (EVs), as one type of membrane-enclosed nanoparticle, are attractive mediators of intercellular and inter-organ communication in various diseases, including HF [46,47,48,49]. Currently, EVs are divided into three types of vesicles in terms of their origin and mechanisms of biogenesis [50] (see Figure 1): (1) Exosomes (EXOs) (50–160 nm); (2) microvesicles (MVs) (100 nm–1000 nm) and (3) apoptotic bodies (APO-EVs) (1–5 μm). EXOs are a type of EV with an endosomal origin made by sequential invagination of the cell membrane resulting in the formation of multivesicular bodies (MVBs). The MVBs will ultimately generate exosomes by fusing with the plasma membrane and undergoing exocytosis [51,52]. Increasing evidence suggests that the biogenesis of EXOs is tightly regulated not only by the endosomal sorting complex required for transport (ESCRT) machinery, syndecan-syntenin-ALIX (Apoptosis-linked gene 2-interacting protein X), tetraspanins and ceramides during the formation of MVBs, but also by cytoskeletal elements, molecular motors and Ras-associated binding GTPases (RABs) during the transport/docking of MVBs from the cytoplasm to cell membrane. In addition, the final fusion of MVBs with the plasma membrane and the secretion of EXOs are driven by the soluble N-ethylmaleimide-sensitive factor attachment protein receptors (SNAREs), sGTPases, and calcium [50,52].

MVs are membrane-bound vesicles with a size range of 100 nm to 1000 nm, produced by direct budding and pinching of the plasma membrane. Although the biogenesis mechanisms are not as well understood, increasing evidence suggests that the formation of MVs is highly related to the regulation of cytoskeletal elements by small GTPases, such as Rho (RAS homolog) family and ADP-ribosylation factors (ARFs). They are also associated with the recruitment of the tumor susceptibility gene 101 (TSG101) by arrestin domain-containing protein 1 (ARRDC1) to the plasma membrane, facilitating the shedding and release of MVs [53,54].

Other than EXOs and MVs, apoptotic bodies are also EVs generated by dying cells during apoptosis and are generally recognized and engulfed by phagocytes [55]. However, recent studies have suggested that apoptotic bodies are generated by budding from the plasma membrane. The formation of apoptotic bodies is regulated by apoptotic cell disassembly via several molecular regulators, including ROCK, Pannexin-1, and Plexin-B2 (well-summarized in [56]). In addition, another novel beads-on-a-string membrane structure is also involved in the formation of apoptotic bodies [57]. Increasing evidence suggests that apoptotic bodies not only facilitate the clearance and degradation of apoptotic materials but also contain other biomolecular cargos (e.g., miRNAs, DNA, protein, and lipids) to mediate intercellular communication [58,59].

Although these membrane-bound vesicles differ in size distribution and mechanism of biogenesis, they are secreted by cells under various pathophysiological conditions into the extracellular space, where they mediate intercellular and inter-organ communication via EV bioactive substances, including proteins, RNAs, DNAs, and lipids. Accumulating evidence suggests that the packaging of cargos into EVs is highly selective and tightly regulated. In this review, we will also focus on the selective packaging of miRNAs into EVs.

### 2.2. Mechanisms of miRNA Selection into EVs

Extracellular miRNAs (Ex-miRs) have been found to be stably transported by ribonucleoproteins (RNPs), lipoproteins, and neutrophil extracellular traps [60,61,62]. However, it has become attractive to view EV function as intercellular and inter-organ mediators conveying their cargos, including noncoding RNAs, in particular, miRNAs. Interestingly, the existence of EV-miRs in the circulation of patients with cardiovascular diseases (CVDs), including HF, has raised the possibility that EV-miRs serve as prognostic and diagnostic markers and potential therapeutic targets [63,64,65]. However, the underlying mechanisms by which miRNAs are selectively secreted remain unclear.

In 2010, a study by Kosaka et al. [21] revealed that secretory miRNAs are regulated by the neutral sphingomyelinase 2 (nSMase 2), which regulates the biogenesis of ceramide and triggers exosome secretion, rather than by the ESCRT system, while the members of ESCRT regulate the biogenesis of EXOs. Increasing studies further suggest that the sorting of miRNA into EVs and their secretion are controlled by specific sequence motifs present in miRNAs that are recognized by RNA-binding Proteins (Figure 1-VIII), including heterogeneous nuclear ribonucleoprotein A2B1 (hnRNPA2B1) [66] and hnRNPU [67], Y-box protein 1 (YBX1) [68,69], synaptotagmin-binding cytoplasmic RNA-interacting protein (SYNCRIP) [70,71], ELAV-like protein 1 (or HuR, human antigen R) [72,73], HSP90AB1, XPO5 and major vault protein (MVP) [72,74], and serine/arginine splicing factor 1 (SRSF1) [75]. Recently, another RNA-binding protein, Lupus La, has been identified to mediate the selective sorting of miRNAs into EVs, and in particular, it selectively sorts miRNA-122 through the specific motifs located at the 3′ end of miRNA-122 [76]. Further studies by analyzing either the EV-enriched (EXOmotifs) or cell-enriched miRNA sequence (CELLmotifs) and structure revealed that the sorting sequences (EXOmotifs) present in miRNAs determine their secretion by EVs. Two novel RNA-binding proteins, Alyref and Fus, have also been identified to function as at least two RNA-binding proteins responsible for EXOmotif recognition and miRNA export into EVs [19].

Additionally, the post-translational modifications of some RNA-binding proteins, such as SUMOylation [66], oligomerization and ubiquitination [73], liquid-liquid phase separation (LLPS) [77], and O-GlcNAcylation [78], also control the sorting of miRNAs into EVs. Interestingly, cell activation-dependent alterations of miRNA targets promote the sorting of miRNAs into EVs [79], and modifications of miRNAs also determine the distribution of miRNA in EVs, such as 3′-end uridylation rather than adenylation [20]. In addition, the ALG-2-interacting protein X (Alix), an accessory protein of ESCRT, is also involved in the miRNA sorting to EVs by interacting with Ago2 and miRNAs during the EV biogenesis [80]. Caveolin-1 (Cav-1) was identified as the first membrane protein to be involved in the selective sorting of miRNAs to EVs by directly interacting with hnRNPA2B1, and the phosphorylation of Cav-1 at Y14 not only promotes the O-GlcNAcylation of hnRNPA2B1 but also enhances the interaction between Cav-1 and O-GlcNAcylated hnRNPA2B1, subsequently facilitating the trafficking of the Cav-1/hnRNPA2B1/miRNAs complex into MVs [78]. The studies and mechanisms cited above strongly support the conclusion that the sorting of miRNAs into EVs is highly selective and that EV secretion is tightly controlled, which may contribute to pathophysiological alterations in human diseases, including HF. For example, these EV sorting mechanisms of miRNAs have been potentially involved in the biogenesis of EV-miRNAs in heart failure (See Table 1).

## 3. EV miRNAs in the Pathogenesis of Heart Failure

To better understand the role of EV-miRNAs in cardiac homeostasis and pathology, recent large-scale single-cell sequencing data revealed that the cellular composition of the adult human heart is heterogeneous and shows transcriptional and cellular diversity [91,92]. The human heart is composed of nine major cell types, including cardiomyocytes, cardiac fibroblasts, endothelial cells, macrophages, etc., and 20 sub-clusters of cell types within the heart [91]. Cardiac homeostasis is normally maintained by dynamic cell-cell and cell–extracellular matrix interactions [93]. When this cardiac homeostasis is disrupted, such as in response to cardiac disease or cardiac damage, the heart will undergo remodeling, which includes fibrosis and hypertrophy—both hallmarks of the HF state [94,95]. Extracellular vesicles are involved in both intercellular and inter-organ crosstalk via the effective transfer of bioactive substances into recipient cells, in particular miRNAs, which are key players in the pathogenesis of HF [96,97,98] (See Table 2, and as illustrated in Figure 2).

### 3.1. EV-miRNA in the Cardiac Hypertrophy

Under cardiac stress, the heart undergoes left ventricular hypertrophy, which is an initial compensatory mechanism. Although cardiac fibroblast-derived EVs play a critical role in inducing cardiac hypertrophy by activating the renin-angiotensin system in cardiomyocytes [121], the EV-enriched miRNAs secreted by cardiac fibroblasts in response to cardiac stress, such as miRNA-21-3p [41] and miRNA-27a-5p [42], can also be taken up by cardiomyocytes, resulting in the cardiac hypertrophy via translational inhibition of both SORBS2 and PDLIM5 [41] or PDLIM5 [42]. In addition, studies from our group also suggest that in response to cardiac stress, cardiac fibroblasts secrete EVs abundant with three miRNAs, including miRNA-27a, miRNA-28-3p, and miRNA-34a. They are taken up by cardiomyocytes, where they mediate the oxidative stress response by targeting Nrf2/ARE signaling, leading to cardiac hypertrophy [99]. In addition to cardiac fibroblasts, adipocytes increase miR-200a expression and secretion by EVs in response to selective activation of PPARγ signaling, and the uptake of EV-miR-200a by cardiomyocytes results in decreased tuberous sclerosis complex (TSC1) and subsequent mTOR activation, leading to cardiomyocyte hypertrophy [100].

### 3.2. EV-miRNA in the Cardiac Fibrosis

Cardiac fibrosis is another hallmark of chronic HF, characterized by extracellular matrix degradation and collagen accumulation. As one of the cardiac-specific miRNAs, miRNA-208a can also be secreted via EVs to mediate the intercellular communication between cardiomyocytes and cardiac fibroblasts and further facilitate cardiac fibroblast proliferation and differentiation into myofibroblasts [104], being transcribed by an intron of alpha-myosin heavy chain (Myh6) in cardiomyocytes, and contributes to cardiac hypertrophy and conduction defects in the heart [101,102]. This molecular mechanism further revealed that EV-miRNA-208a promotes cardiac fibrosis by targeting Dyrk2 (dual-specificity tyrosine phosphorylation-regulated kinase 2) to promote NFAT (nuclear factor of activated T cells) dephosphorylation and nuclear translocation, which triggers fibrosis. In addition, miRNA-217 was found to be elevated in the hearts of patients with chronic HF, and exogenous overexpression of miRNA-217 in cardiomyocytes in the thoracic aortic constriction (TAC)-induced HF model demonstrated enhanced pressure overload-induced cardiac dysfunction and cardiac remodeling (cardiac fibrosis and hypertrophy). Interestingly, miRNA-217 not only directly regulates cardiac hypertrophy but also indirectly contributes to cardiac fibrosis via cardiomyocyte-derived EVs by targeting PTEN in both cells [103]. Moreover, under cardiac stress, such as mechanical stretch and pressure overload, cardiomyocyte-specific Peli1 (Pellino E3 Ubiquitin Protein Ligase 1) has been involved in the enhanced transcription of miRNA-494-3p in cardiomyocytes via regulating NF-ĸB/AP-1 activation, and promotes the secretion of EV-enriched miRNA-494-3p into cardiac fibroblasts to activate cardiac fibroblasts by targeting PETN to enhance the phosphorylation of AKT, ERK, and SMAD2/3 [105]. Furthermore, HF patients with familial dilated cardiomyopathy also demonstrate late-stage cardiac fibrosis. A recent study using human iPSCs-derived cardiomyocytes as a disease model suggested that EVs derived from Ang II-stimulated DCM highly exhibit increased cardiac fibrosis and impaired cardiac function in vitro and in vivo compared to control cardiomyocytes via EV-enriched miRNA-218-5p targeting TNFAIP3 to activate TGF-β signaling [106]. Other than cardiomyocytes, Ang II-stimulated adipocytes can also secrete EV-enriched miRNA-23a-3p, which transforms cardiac fibroblasts into myofibroblasts and promotes collagen accumulation by targeting RAP1 (Ras-related protein 1) [107]. In addition, cardiac infiltration of CD4^+^ T cells is implicated in the healing process post-myocardial infarction (MI), contributing to cardiac fibrosis and dysfunction [122]. A mechanistic study further revealed that cardiac-activated CD4+ T cells can transport excessive miR-142-3p via EVs into cardiac fibroblasts in which miR-143-3p activates the WNT signaling pathway by targeting APC (adenomatous polyposis coli) to transform cardiac fibroblasts into activated myofibroblasts boosting post-ischemic ventricular remodeling in the progression of HF [108]. It has been well-documented that miRNA-21 is a central regulator of cardiac fibrosis in HF and shows therapeutic potential as a target for HF treatment [109,110,111,112]. Recently, using the TAC-induced HF model combined with single-cell sequencing, bioinformatics analyses revealed that EV-enriched miR-21 also determines macrophage-fibroblast crosstalk and promotes the transition from cardiac fibroblasts to activated myofibroblasts, leading to cardiac fibrosis [113].

### 3.3. EV-miRNA in Cardiac Angiogenesis during Heart Failure

In response to cardiac injury, intercellular communication is fundamental for maintaining homeostasis and integrity. The aforementioned studies have demonstrated that EV-enriched miRNAs derived from other types of cardiac cells, including macrophages, T cells, and adipocytes, contribute to cardiac fibrosis and/or hypertrophy in HF. EVs secreted from cardiomyocytes also play an important role in anti-angiogenesis through miRNA transfer in HF. Recently, miRNA-200c-3p has been found to be one of the most enriched miRNAs in hypertrophic cardiomyocyte-derived EVs under pressure overload and was transported into endothelial cells (ECs) in which it functions as a detrimental anti-angiogenic factor to impair endothelial function including proliferation, migration, and tube formation [114]. It has also been shown that hypertrophic cardiomyocytes induced by Ang II also release miRNA-29a via EVs to inhibit the proliferation, migration, and angiogenic ability of cardiac microvascular ECs [115]. Other than hypertrophic cardiomyocytes, activated cardiac fibroblasts can also secrete EVs abundant with miRNA-200a-3p, which mediates the cross-talk between cardiac endothelial cells and induces endothelial dysfunction by targeting the ETS1/VEGF-A signaling axis [116]. A recent study also suggested that M1 macrophages secrete pro-inflammatory EVs post-MI, which exert anti-angiogenic effects by transferring EV-enriched miR-155 into cardiac ECs to inhibit the angiogenesis and further accelerate MI injury by targeting the Sirt1/AMPKα2-endothelial nitric oxide synthase and RAC1-PAK2 signaling pathways [117].

As discussed above, cardiac EV-enriched miRNA-mediated intercellular communications play an important role in the pathogenesis of HF. However, recent studies [49,118] highlight the possibility that cardiac-derived EVs contribute to inter-organ communication during the progression of HF.

### 3.4. Cardiac EV miRNA-Mediated Inter-Organ Communication in Heart Failure

Increasing evidence suggests that heart-brain communication at the miRNA level contributes to neuronal dysfunction in the brain mediated by EVs in the HF state [49,118]. miRNA-1 is a cardiac miRNA abundantly expressed in the myocardium [102]. Circulating miRNA-1 was significantly increased in patients with acute myocardial infarction (AMI) [102,123]. Interestingly, cardiac-derived miRNA-1 can be transported from the infarcted heart into the hippocampus via EVs, where cardiac miRNA-1 causes neuronal microtubular damage. This is independent of brain hypoperfusion induced by MI [118], suggesting a novel mechanism by which the damaged heart contributes to brain dysfunction. A study from our group also demonstrated that cardiac-derived miRNAs can be sorted into EVs and circulate into the rostral ventrolateral medulla (RVLM) of the brain stem, where these miRNAs evoke sympathetic excitation by targeting Nrf2/ARE signaling to induce oxidative stress [49]. Moreover, it is well-documented that neutrophils promote the progression of acute MI by releasing ROS, granular components, and extracellular traps to aggravate inflammation [124,125]. Clinical evidence also shows that the numbers of peripheral blood neutrophils and plasma EVs correlate closely with the extent of acute MI, including infarct size, mortality, and HF development [48,126,127]. Interestingly, miRNA analysis of human plasma EVs from patients with AMI demonstrated twelve significantly enriched miRNAs, and two out of twelve, including miRNA-126-3p and -5p, were highly regulated and secreted by ECs post- acute MI and are responsible for cell adhesion and chemotaxis [119]. Importantly, recent studies from the same group revealed that cardiac ECs in response to cardiac injury rapidly release VCAM-1^+^ EVs containing miRNA-126, which preferentially mediate heart-spleen communication. These VCAM-1^+^ EVs rapidly and selectively recruit splenic neutrophils to peripheral blood following myocardial injury. Mechanistic studies further suggest that VCAM-1^+^ EVs-enriched miRNA-126 induces the transcriptional activation of neutrophils in the spleen before they arrive at the ischemic myocardium, contributing to local inflammation and chemokine production [48,120]. In addition, genetic deletion of VCAM-1 from EVs by the CRISPR-Cas9 system or silencing miRNA-126 in vivo by antagomir significantly blocked the mobilization of splenic neutrophils to the ischemic myocardium and reduced myocardial infarct size in a LAD ligation mouse model. This suggests that specific surface proteins such as VCAM-1 in EVs will determine their organotropism and functionalities of EVs.

## 4. Extracellular Vesicle miRNA-Based Prognosis, Diagnosis and Therapeutics of Heart Failure

Although natriuretic peptides and cardiac troponins are currently the most widely employed biomarkers for the prognosis and diagnosis of HF, next-generation biomarkers, including soluble source of tumorigenicity 2 (sST2), proenkephalin, growth differentiation factor-15 (GDF-15), and galectin-3 (Gal-3) have also been well-established as promising biomarkers for HF diagnosis and prognosis [128,129]. The discovery of altered levels of circulating miRNAs in patients with HF provides the possibility to utilize circulating miRNAs, in particular, EV-enriched miRNAs, as biomarkers for HF [63,64,130] (See Table 3).

Recently, circulating EV-miRNAs have been emerging as non-invasive prognostic and diagnostic biomarkers for HF, and their potential clinical applications have been well-summarized [141]. For example, some circulating EV-miRNAs, including miR-92-5p, miR-146a, miR-181c, and miR-495, showed diagnostic potential for HF, and other EV-enriched miRNAs, such as miR-192, miR-34a, miR-425, and miR-744, represent promising prognostic biomarkers for HF. Moreover, two circulating EV-enriched miRNAs associated with HF, including miR-30d-5p and miR-126a-5p, have shown promising potential as biomarkers for HFpEF (heart failure with preserved ejection fraction) in diabetes mellitus because their down-regulations in circulating EVs and the left ventricle remain consistently correlated with decreased cardiac output [142]. In addition, circulating miR-30d, especially EV-miR-30d, has shown high potential as a biomarker to evaluate left ventricular remodeling and clinical outcomes for patients with HF [140,143].

Other than serving as prognostic and diagnostic biomarkers, EV-miRNAs have been used as therapeutic targets for HF (See Table 4). Some miRNAs such as miR-126, miR-146a, miR-125a-5p, miR-125b-5p, miR-29b, miR-98-5p, miR-30e, and miR-30d have shown protective effects on cardiac function. Extracellular vesicle-based delivery of these miRNAs shows promising clinical applications in HF therapy [144,145,146,147,148,149,150]. A recent study showed the beneficial effects of adipose tissue-derived mesenchymal stem cell (ADSC)-derived EVs on acute MI-induced cardiac injury by EV-enriched miR-205, suggesting a promising therapeutic potential [151]. Moreover, increasing evidence suggests that EVs secreted by other stem cells, including human cardiac progenitors (hCPCs), mesenchymal stem cells (MSCs), and induced pluripotent stem cells (iPSCs), demonstrate angiogenic and cardioprotective properties in MI rodent models by significantly increasing the proliferation, migration, and tube formation of endothelial cells [152,153,154]. EV miRNA profiling further revealed that some angiogenic and cardioprotective miRNAs, such as miR-210 [152,153,154], miR-126, and miR-17-92 [154], are highly enriched in these stem cell-derived EVs. For example, silencing the miR-210 of MSC-EVs significantly impaired the pro-angiogenic effects in vitro and in a mouse MI model [153]. Recently, human umbilical cord mesenchymal stem cell-derived EVs pre-loaded with miR-29b mimics also demonstrated effective antifibrotic activity to prevent excessive cardiac fibrosis post-MI [146]. These studies suggest that endogenous miRNAs enriched in stem cell-derived EVs or exogenous miRNA mimics pre-loaded into stem cell-EVs exhibit a potential for MI therapy. Although stem cell-derived EV-miRNAs have shown promising therapeutic potential for HF management, it remains questionable if the beneficial contributions of other EV components, such as proteins, lipids, and various metabolites, affect cardiac function in the treatment of HF. Moreover, an interesting new therapeutic direction relates to the effects of stem cell-derived EVs in HF by engineering EVs with cardiac homing peptides and using genetic therapeutics for the delivery of miRNA-21 to effectively restore cardiac function after MI [155]. For example, hypoxia-conditioned BM-MSCs (bone marrow mesenchymal stem cells) secreted EVs abundant with miRNA-125b-5p. A cardioprotective miRNA demonstrated high specificity to the ischemic myocardium in a mouse model of acute MI when miR-125-enriched EVs were conjugated with an ischemic myocardium-targeted peptide and then administered by intravenous injection [144]. Alternatively, targeting injured cardiac-secreted EVs by exogenously pre-loading miRNA inhibitors (i.e., antagomirs) into EVs, such as miR-27a, miR-28, and miR-34a [49], as well as miR-126 [48], may represent another therapeutic strategy.

## 5. Perspectives and Future Directions

Extracellular vesicle miRNAs are attractive candidates as diagnostic and prognostic biomarkers as well as therapeutic targets. However, using EV-miRNAs for these purposes still faces great challenges in becoming well-accepted biomarkers for prognosis and diagnosis of HF. First, rigorous standards for performing EV research are imperative and are time-consuming processes, including EV categorization, purification, and separation technology development and characterization. Uniform standard guidelines are urgently needed in the field. Second, the pathogenesis of HF is complicated, and cardiac function is also influenced by systemic communication with other organs. Selective criteria are needed to evaluate the potential of EV-miRNAs for use as diagnostic and prognostic biomarkers for HF. In addition, the cell origins and categories of circulating EVs are heterogeneous, and the exploration of cell origin and sorting mechanisms of circulating EV-miRNAs will be of interest to better understand the pathophysiological functions of circulating EV-miRNAs in HF. Third, the sample size used for evaluating circulating EV-miRNAs as biomarkers for HF is still relatively small, which is insufficient to support their use as sensitive biomarkers at this time.

Although EVs have exhibited excellent properties and functions, including regulatory ability, physical stability, and immunogenicity, which render EVs a potential novel platform for drug delivery and precision therapy, the targeting specificity and delivery efficiency remain challenges. A cardiac homing peptide-guided delivery system has been used for the treatment of MI [159,160]. Given that cardiac-derived EVs also contribute to neuronal dysfunction in the brain in HF [49,118], engineered EVs derived from dendritic cells expressing Lamp2b (an exosomal membrane protein), fused to the neuron-specific RVG peptide [161], may be therapeutically used for HF management. In addition, the next generation of “Hybrid EVs” [162] produced by stem cell-derived EVs fused with modified liposomes may represent an innovative therapeutic strategy for HF treatment. The resulting hybrid EVs will not only possess the intrinsic immunomodulatory effects and blood barrier penetrating ability of original stem cell-derived EVs but also highly enrich some cardioprotective miRNAs by pre-loading them into parental liposomes. Importantly, the parental partners can be modified by either engineering them with organotropic peptides or integrating multimodal imaging capability for clinical tracking, guiding, and optimizing. Taken together, miRNAs are differentially regulated in response to heart injury and selectively sorted into EVs, contributing to the pathogenesis of HF via intercellular and/or inter-organ communications, which makes it possible to become a promising type of prognostic and diagnostic biomarker and therapeutic target for HF.

## Figures and Tables

**Figure 1 cells-12-02145-f001:**
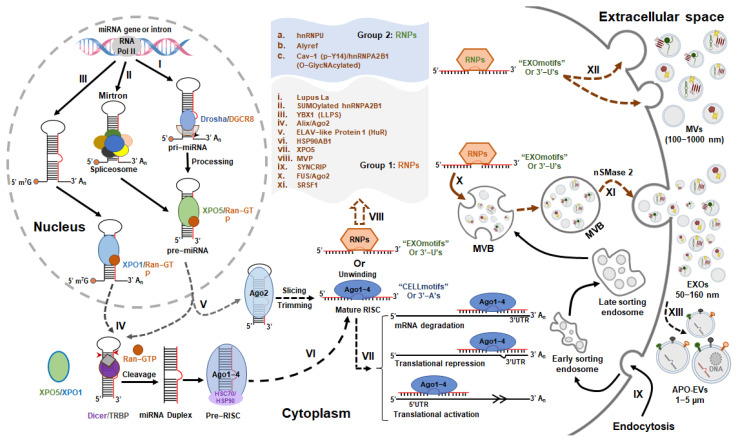
The biogenesis and potential sorting mechanisms of EV-miRNAs. The biogenesis of miRNAs is regulated by canonical (I) and non-canonical pathways (II and III), respectively. The latter is composed of Drosha/DGCR8-independent (II and III), Dicer-dependent (IV) and Dicer−independent pathways (V); The sorting sequences or 3′ modifications of miRNAs determine their incorporation in EVs (EXOmotifs or 3′−Uridylation) or cellular retention (CELLmotifs or 3′−Adenylation) [19,20]. The miRNAs with CELLmotifs or 3′−A’s are combined with AGO proteins to form the mature RISC (VI), and then mediate either the mRNA degradation or translational repression by binding to 3′ UTR, and also regulate the translational activation by binding to 5′ UTR (VII); The miRNAs with EXOmotifs or 3′−U’s are specifically recognized by RNA binding proteins (RNPs) (VIII), and are then sorted into microvesicles (MVs) by binding with group 2 RNPs (XII), or into Exosomes (EXOs) by binding with group 1 RNPs (X). In addition, the secretion of miRNAs is also controlled by neutral sphingomyelinase 2 (nSMase2) (XI) [21]. IX indicates the endocytosis process; XIII indicates the formation of apoptotic extracellular vesicles (APO-EVs); ALG-2-interacting protein X (Alix); Caveolin-1 (Cav-1); DiGeorge syndrome critical region 8 (DGCR8); ELAV-like protein 1 (or HuR, human antigen R); exosomes (EXOs); exportin 1 (XPO1); exportin 5 (XPO5); heat shock protein HSP 90-beta (HSP90AB1); heterogeneous nuclear ribonucleoprotein A2B1 (hnRNPA2B1); heterogeneous nuclear ribonucleoprotein U (hnRNPU); HIV-1 transactivating response (TAR) RNA-binding protein (TRBP); liquid-liquid phase separation (LLPS); major vault protein (MVP); microvesicles (MVs); multivesicular bodies (MVBs); Ras-related nuclear protein (RAN); RNA binding proteins (RNPs); RNA-induced silencing complex (RISC); RNA polymerase II (Pol II); serine/arginine-rich splicing factor 1 (SRSF1); synaptotagmin-binding cytoplasmic RNA-interacting protein (SYNCRIP); Y-box protein 1 (YBX1).

**Figure 2 cells-12-02145-f002:**
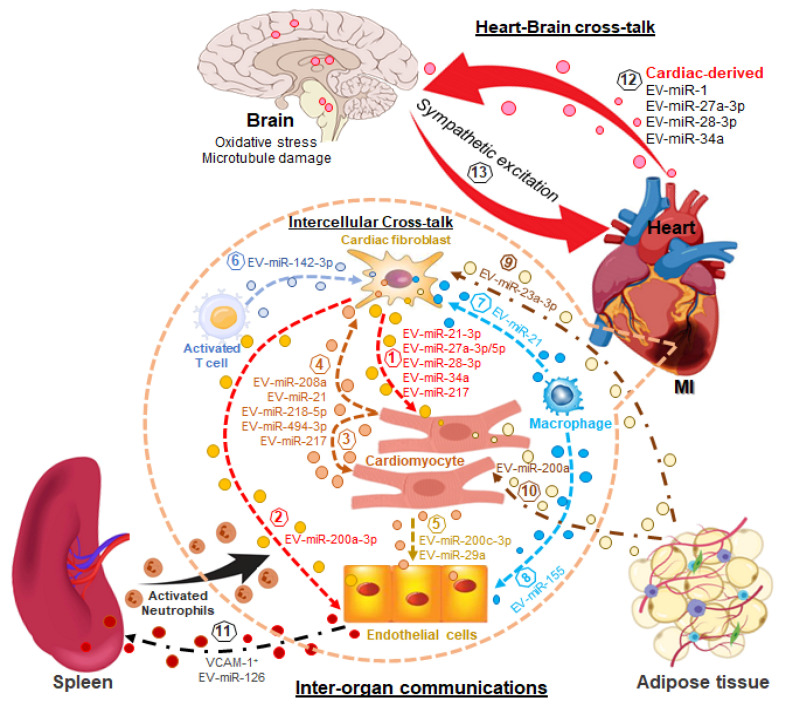
EV miRNAs in the pathogenesis of heart failure via intercellular and inter-organ communication. miRNA-enriched EVs mediate intercellular communications: EVs from cardiac fibroblasts contribute to cardiac hypertrophy (1) and the impairment of angiogenic capacity (2); EVs derived from cardiomyocytes contribute to cardiac hypertrophy (3), fibrosis (4), and the impairment of endothelial proliferation, migration, and tube formation (5); EVs secreted by activated cardiac T cells (6) and macrophages (7) mediate cardiac fibrosis, and the inhibition of angiogenesis (8); miRNA-enriched EVs mediate inter-organ communications between adipose tissue (adipocytes) and cardiomyocytes and fibroblasts leading to fibrosis (9) and hypertrophy (10); heart (cardiac endothelial cells) and spleen (neutrophils) recruiting activated neutrophils to the ischemic region following MI increasing inflammation and promoting myocardial injury (11); cardiac derived cells communicate with neurons in the brain increasing oxidative stress and/or microtubule damage (12) eliciting sympathetic excitation which negatively regulates cardiac function in the setting of heart failure (13).

**Table 1 cells-12-02145-t001:** The potential EV-sorting mechanisms of some HF-associated miRNAs.

miRNAs	Sorting Mechanism	Functions in HF	Ref.
miR-122	The binding of Lupus La protein, hnRNPU and/or HuR to miR122 controls extracellular export	Promote apoptosis, inflammation, fibrosis, pathological hypertrophy and remodeling	[73,76,81,82]
miR-223	Selective sorting of miR-223 into EXOs by phase-separated YBX1 condensates	Promote cardiac fibrosis and hypertrophy	[69,77,83,84]
miR-34c-5p	The binding of Alyref and/or Fus to the CGGGAG motif at the 3′ end of miR-34c	Cardiac hypertrophy	[19,85]
miR-26a	The binding of Alyref and/or Fus to the CGGGAG motif at the 3′ end of miR-34c; alternatively, 3′-end uridylation of miR-26a	Protects the heart against hypertension-induced myocardial fibrosis	[19,20,86]
miR-30c-5p	The binding of hnRNPU to the AAMRUGCU motif of miR-30c-5p	Protects against myocardial ischemia/reperfusion injury	[67,87]
miR-17/92	The binding of cav-1/hnRNPA2B1 complex to miR-17/92 regulates its MV sorting	Hypertrophic and arrhythmogenic cardiomyopathy	[78,88]
miR-1246	The binding of SRSF1 to miR-1246 regulates its exosomal enrichment	Upregulated in diastolic dysfunction	[75,89]
miR-1231	The binding hnRNPA2B1 to the GGAG EXOmotif at the 3′ end of miR-1231	Induction of arrhythmias in ischemic hearts	[70,90]

**Table 2 cells-12-02145-t002:** The EV-miRNAs in the pathogenesis of HF.

Pathological Phenotype	miRNA	Cell Source	Target Cell	Potential Functional Mechanism	Ref.
Cardiac hypertrophy	miRNA-21-3p	CF	CM	Translational inhibition of both SORBS2 and PDLIM5	[41]
	miRNA-27a-5p	CF	CM	Translational inhibition of PDLIM5	[42]
	miRNA-27a-3p, miRNA-28-3p miRNA-34a	CF	CM	Dysregulation of Nrf2/ARE signaling and oxidative stress	[99]
	miR-200a	Adipocyte	CM	Selective activation of PPARγ signaling and decreased TSC1 and subsequent mTOR activation	[100]
	miRNA-208a	CM	CM	Repression of Thrap1 and myostatin expression	[101,102]
	miRNA-217	CF	CM	Targeting PTEN	[103]
Cardiac fibrosis	miRNA-208a	CM	CF	Targeting Dyrk2 to promote NFAT dephosphorylation and nuclear translocation	[104]
	miRNA-217	CM	CF	Targeting PTEN	[103]
	miRNA-494-3p	CM	CF	Targeting PETN to enhance the phosphorylation of AKT, ERK, and SMAD2/3	[105]
	miRNA-218-5p	CM	CF	Targeting TNFAIP3 to activate TGF-β signaling	[106]
	miRNA-23a-3p	Adipocyte	CF	Targeting RAP1	[107]
	miR-142-3p	Activated CD4^+^ T cell	CF	Targeting APC to activate the WNT signaling pathway	[108]
	miRNA-21	MP and/or CM	CF	Targeting Spry1 to augment ERK-MAP kinase activity	[109,110,111,112,113]
Angiogenesis	miRNA-200c-3p	CM	EC	Impaired endothelial migration and tube formation, as well as a lower proliferation capacity	[114]
	miRNA-29a	CM	EC	Inhibiting the proliferation, migration, and angiogenic ability of cardiac microvascular ECs	[115]
	miRNA-200a-3p	Activated CF	EC	Targeting ETS1/VEGF-A signaling axis	[116]
	miRNA-155	Activated MP	EC	Targeting Sirt1/AMPKα2 and RAC1–PAK2 signaling pathways	[117]
Inter-organ communications	miRNA-1	CM	Neuron	Targeting TPPP/p25 to disturb the stability of neuronal microtubules	[102,118]
	miRNA-27a-3p, miRNA-28-3p and miRNA-34a	CM and/or CF	Neuron	Targeting Nrf2/ARE signaling to induce oxidative stress and subsequently elicit sympathetic excitation	[49]
	miRNA-126	EC	NEUT	Transcriptional activation of NEUTs and contribution to cardiac inflammation and chemokine production	[48,119,120]

Abbreviations: adenomatous polyposis coli (APC), cardiac fibroblast (CF), cardiac myocyte (CM), dual-specificity tyrosine phosphorylation-regulated kinase 2 (Dyrk2), endothelial cell (EC), ETS Proto-Oncogene 1 (ETS1), macrophages (MP), neutrophils (NEUTs), nuclear factor of activated T cells (NFAT), Ras-related protein 1 (RAP1), Sprouty homolog 1 (Spry1), thyroid hormone-associated protein 1 (Thrap1), tuberous sclerosis complex (TSC1), tubulin polymerization promoting protein (TPPP/p25).

**Table 3 cells-12-02145-t003:** The significance of EV-miRNAs as diagnostic and prognostic biomarkers in heart failure.

miRNA	Biomarker Type	Regulation in HF	Source of miRNAs	Cohort Size	Analysis Method	Ref.
miR-92-5p	Diagnostic	Up	Serum (H)	*n* = 28	qRT-PCR	[131]
miR-146a		Up	Plasma (H)	*n* = 192	qRT-PCR	[132,133]
miR-181c		Up	Serum (H)	*n* = 57	qRT-PCR	[134]
miR-495		Up	Plasma (D)	*n* = 11	qRT-PCR	[135]
miR-192	Prognostic	Up	Plasma (H)	*n* = 91	qRT-PCR	[136]
miR-34a		Up	Plasma (H)	*n* = 359	qRT-PCR	[137]
miR-194		Up	Serum (H)	*n* = 21	qRT-PCR	[138]
miR-425		Down	Serum (H)	*n* = 31	qRT-PCR	[139]
miR-744		Down	Serum (H)	*n* = 31	qRT-PCR	[139]
miR-30d	Prognostic for CRT response	Down	Plasma (H)	*n* = 92	qRT-PCR	[140]

Abbreviations: cardiac resynchronization therapy (CRT); human (H); dog (D).

**Table 4 cells-12-02145-t004:** The potential therapeutic applications of stem cell-derived EV miRNAs in heart failure.

miRNA	Sources	Animal Model	Function	Ref.
miR-125b-5p	MSC-derived hypo-EVs	MI	Suppress the expression of the pro-apoptotic genes p53 and BAK1 in cardiomyocytes	[144]
miR-98-5p	hypoxic BMMSCs	I/R	Targeting TLR4 and the PI3K/Akt signaling pathway	[145]
miR-29b	Exogenously loaded	MI	Antifibrotic activity to prevent excessive cardiac fibrosis	[146]
miR-129-5p	MSCs	MI	Targeting TRAF3 and the following NF-κB signaling	[147]
miR-126	ADSC	AMI	Protecting cardiac cells from apoptosis, inflammation, fibrosis, and increased angiogenesis.	[156]
miR-146a	ADSCs	AMI	Targeting EGR1 to attenuate AMI-induced myocardial damage	[157]
miR-125a-5p	MSCs	I/R	Increase M2 macrophage polarization, promote angiogenesis, and attenuate fibroblast proliferation and activation	[150]
miR-205	ADSC	MI	Promote the proliferation and migration of ECs, facilitate angiogenesis, and inhibit cardiomyocyte apoptosis	[151]
miRNA-21	Exogenously loaded	MI	Reduce the PDCD4 expression and attenuate cell apoptosis	[155]
miR-30e	MSCs	MI	Inhibit LOX1 expression and impair the NF-κB p65/Cas-9 signaling	[147]
miR-210	MSCs	MI	Targeting Efna3 to improve angiogenesis	[153]
miR-17-92	CPCs	I/R	Inhibit fibrosis	[158]

Abbreviations: adipose tissue-derived mesenchymal stem cell (ADSC); acute myocardial infarction (AMI); bone marrow mesenchymal stem cells (BMMSCs); cardiac progenitor cells (CPCs); Early growth response factor 1 (EGR1); ephrin A3 (Efna3); lectin-like oxidized low-density lipoprotein receptor 1 (LOX1); ischemia-reperfusion (I/R); mesenchymal stem cells (MSCs); myocardial infarction (MI); programmed cell death 4 (PDCD4).

## Data Availability

Not applicable.

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
