# Peer review of "Extracellular Vesicle MicroRNAs in Heart Failure: Pathophysiological Mediators and Therapeutic Targets"

_cells, 2023, doi:10.3390/cells12172145_

Round 1

Reviewer 1 Report

The manuscript by Tian et al. entitled “Extracellular vesicle microRNAs in heart failure: pathophysiological mediators and therapeutic targets” provides a thorough summary of the role of circulating extracellular vesicle (cEV)-containing microRNAs (miRNAs) in heart failure (HF) highlighting their therapeutic potential. The paper is well-written and clear and it covers most of the literature on this topic. I would only suggest (1) adding a schematic figure summarising the role of cEV-containing miRNAs in the pathophysiology of HF. (2) A table listing key miRNAs found as potential diagnostic and prognostic biomarkers would also be desirable. (3) Finally, the section devoted to therapeutic targets should be concisely expanded. 

No comments.

Author Response

Responses to Reviewers:

We thank the editor for this opportunity to revise our manuscript, and appreciate all the constructive suggestions and comments from the reviewers. Changes to the revised manuscript are marked in red text. To further address and clarify the concerns of reviewers, we would like to provide a point-by-point response as follows:

Reviewer #1

The manuscript by Tian et al. entitled “Extracellular vesicle microRNAs in heart failure: pathophysiological mediators and therapeutic targets” provides a thorough summary of the role of circulating extracellular vesicle (cEV)-containing microRNAs (miRNAs) in heart failure (HF) highlighting their therapeutic potential. The paper is well-written and clear and it covers most of the literature on this topic.”

Response: Thank you for your positive comments.

I would only suggest (1) adding a schematic figure summarising the role of cEV-containing miRNAs in the pathophysiology of HF. (2) A table listing key miRNAs found as potential diagnostic and prognostic biomarkers would also be desirable. (3) Finally, the section devoted to therapeutic targets should be concisely expanded.

Response: Thank you for your constructive suggestions and comments. We would like to provide a point-by-point response to these remaining concerns as follows:

Point 1. “adding a schematic figure summarising the role of cEV-containing miRNAs in the pathophysiology of HF.”

Response: We have added figure 2 and its legend as suggested on page 6, lines 221-245, which has been highlighted in red. Thank you for this important comment.

Point 2. “A table listing key miRNAs found as potential diagnostic and prognostic biomarkers would also be desirable”.

Response: We have provided this table (Table 3) as suggested on page 10-11, lines 378-382 and highlighted these revisions in red.

Point 3. “Finally, the section devoted to therapeutic targets should be concisely expanded.

Response: Thank you for this suggestion. We have separated the section on stem cell-derived EV and miRNA from the "future directions" and expanded the discussion in this section to discuss therapeutic targets, please see pages 11-12, lines 397-438, and page 12 lines 456-459. 

Reviewer 2 Report

The review by Tian et al provides a comprehensive overview of our understanding of the role of extracellular vesicles and their microRNA cargo in heart failure.

A few minor suggestions to help the flow

to include a Table 3 outlining the miRNA that have been identified with potential for diagnosis and therapy (Section 4 of the manuscript)

Separate the section on stem cell-derived EV and miRNA so that it is not within the "future directions" final summary

Author Response

Responses to Reviewers:

We appreciate all the constructive suggestions and comments from the reviewers. Changes to the revised manuscript are marked in red text. To further address and clarify the concerns of reviewers, we would like to provide a point-by-point response as follows:

Reviewer #2

The review by Tian et al provides a comprehensive overview of our understanding of the role of extracellular vesicles and their microRNA cargo in heart failure. A few minor suggestions to help the flow to include a Table 3 outlining the miRNA that have been identified with potential for diagnosis and therapy. Section 4 of the manuscript Separate the section on stem cell-derived EV and miRNA so that it is not within the "future directions" final summary

Response: Thank you for your constructive suggestions and comments.

First, we have provided table 3 (list of EV-miRNAs for diagnostic and prognostic biomarkers) as suggested on pages 10-11, and table 4 (list of EV-miRNAs for potential therapeutic applications in heart failure) on page 12, and highlighted these revisions in red.

Second, we also separated the section on stem cell-derived EV and miRNA from the "future directions" and expanded the discussion in this section to discuss therapeutic targets, please see pages 11-12, lines 397-438, and page 12 lines 456-459. 

Round 2

Reviewer 1 Report

Congratulations on your work! 

No comments.